# Identification of Bridge Key Performance Indicators Using Survival Analysis for Future Network-Wide Structural Health Monitoring

**DOI:** 10.3390/s20236894

**Published:** 2020-12-02

**Authors:** Nicola-Ann Stevens, Myra Lydon, Adele H. Marshall, Su Taylor

**Affiliations:** 1School of Natural and Built Environment, Queen’s University Belfast, David Keir Building, Belfast BT9 5AG, UK; nstevens01@qub.ac.uk (N.-A.S.); s.e.taylor@qub.ac.uk (S.T.); 2School of Mathematics and Physics, Queen’s University Belfast, University Rd, Belfast BT7 1NN, UK; a.h.marshall@qub.ac.uk

**Keywords:** structural health monitoring, bridge management systems, survival analysis, Markov chains

## Abstract

Machine learning and statistical approaches have transformed the management of infrastructure systems such as water, energy and modern transport networks. Artificial Intelligence-based solutions allow asset owners to predict future performance and optimize maintenance routines through the use of historic performance and real-time sensor data. The industrial adoption of such methods has been limited in the management of bridges within aging transport networks. Predictive maintenance at bridge network level is particularly complex due to the considerable level of heterogeneity encompassed across various bridge types and functions. This paper reviews some of the main approaches in bridge predictive maintenance modeling and outlines the challenges in their adaptation to the future network-wide management of bridges. Survival analysis techniques have been successfully applied to predict outcomes from a homogenous data set, such as bridge deck condition. This paper considers the complexities of European road networks in terms of bridge type, function and age to present a novel application of survival analysis based on sparse data obtained from visual inspections. This research is focused on analyzing existing inspection information to establish data foundations, which will pave the way for big data utilization, and inform on key performance indicators for future network-wide structural health monitoring.

## 1. Introduction

The industrial application of Structural Health Monitoring (SHM) systems in the network-wide management of bridges on transport networks is broadly underutilized. Visual inspection remains the most common method for bridge assessment despite the ability of SHM to provide dramatically improved data in terms of detail, accuracy and repeatability. A recent UK study [1] revealed that, with the exception of a few strategically important structures, SHM is currently only deployed on individual structures which have been identified as having significant defects. In general, the data relating to bridge condition are, consistently, not exploited to their full potential and used only in reactive analytics when the condition is very poor rather than used earlier in the bridge lifetime. Engineering judgement is predominantly used to make operational decisions in the overall management of bridges within road networks. A further study in 2019 highlighted that in the UK there is significant variation in the visual inspection methods adopted and therefore the data have inherent inaccuracies [2]. As a result, Bridge Management Systems (BMS) have limited capability to implement Artificial Intelligence (AI)-based solutions under the present condition rating methods and remain dependent upon human input. Recent developments in Population-Based SHM (PBSHM) provide an opportunity for a shift change in the potential inclusion of sensor data for network level bridge assessment. In this case, data obtained from a population of structures can provide extra information and improve damage detection for each individual structure. A normal condition is identified across a population of structures that is robust to environmental variation, allowing damage indicators to be transferred between structures but has only been tested on a homogeneous population of structures [3]. In order to adopt such techniques to a heterogeneous population of structures, there is a need to identify key performance indicators at a network level which would allow for cluster analysis and the establishment of structural similarities. The greatest obstacle to this is the lack of analysis of the historical data. In order to make our bridge inspections more efficient, economical and effective at local and global levels, there is a need to establish baseline data sets [4]. A clear digital framework is needed to enable the integration of sensor data into existing BMS, whereby historic data informs the monitoring needs and provides a beneficial cost–benefit ratio. In advanced cases, a BMS consists of four main components: Firstly, the database where the data on each bridge is stored. This information includes details about the bridge, inspection history and maintenance data. Secondly, a deterioration model is used to predict the future condition of the bridge and its components. Thirdly, a cost model is presented, which estimates costs, associated with maintenance, rehabilitation and replacement. The final component is an optimization model, which is used to determine the strategy with the lowest cost. If fully exploited, the deterioration model has the capability to identify bridge and non-bridge factors which impact deterioration at network level and therefore can highlight potential properties to monitor at PBSHM. This paper does not provide a comprehensive description of all maintenance methods and policies available in the literature. A critical review of some of the main approaches in bridge predictive maintenance modeling is presented and the challenges in their adaptation to the future network-wide management of bridges are outlined. Initially, we discuss the developments relating to Markov-chain theory for bridge deterioration and how it has been adopted in the current state-of-the-art bridge management systems. The review of the literature includes methods to improve the accuracy of the Markov model. While these methods can improve the accuracy of the Markov model, they do not address the assumptions and limitations of the Markov model. This has prompted the introduction of the semi-Markov model which allows some of these assumptions to be relaxed. One of the methods used to calculate semi-Markov transition probabilities was reliability/survival analysis and was first presented [5] as a technique to calculate transition probabilities for the semi-Markov approach. More recently it has been used to predict the service life of bridges and the time in each condition rating. In addition to this, a review of artificial intelligence, machine learning and data mining techniques is undertaken as these methods have been used to address some of the assumptions made by the Markov model. A case study looking at the application of survival analysis to the Northern Ireland road network is presented. This will lead into a discussion that illustrates the challenges of enabling more advanced AI-based solutions for the management of bridge assets in ageing infrastructures networks such as those in Europe and the UK. The steps necessary to overcome uncertainties in existing data are outlined with a view towards moving to an operational BMS with AI-based solutions, which can enable the integration on sensor systems for network-wide monitoring.

## 2. Methodologies Adopted for Current Bridge Predictive Maintenance Methods

### 2.1. Markov Theory and Its Application in the Current State-Of-The-Art Bridge Management Systems

Markov chains are the most commonly used stochastic technique [6] used for predicting the future performance of a bridge and for this reason they are used in many current management systems (BMS) Pontis (version 3 1997 [7]) and BRIDGIT (version 3 1998 [8]).

Markov chains are used in bridge performance prediction by assigning each condition rating to a state in the Markov chain and then calculating the probabilities of transition from one condition rating to another over a predefined time interval. As this model is a first order stochastic process, this results in the probability of moving to another state in a pre-defined time step depending only on the present state of the bridge and not on its past condition. This assumption, which is known as the state independence assumption or memoryless property, can be mathematically defined as below in Equation (1), where $i_{t}$ is the state of the process at time t, P is the probability matrix and the symbol | represents “conditional on” so the below expression represents the probability of any future event given the present state and past events.
(1)$$P\left(X_{t+1}=i_{t+1}|X_{t}=i_{t},X_{t-1}=i_{t-1},\ldots ,X_{1}=i_{1},X_{0}=i_{0}\right)\;=P\left(X_{t+1}=i_{t+1}|X_{t}=i_{t}\right)$$

In practice, the Markovian assumption is made without verification which can lead to the quality of the model being questioned [9]. In addition to the state independence assumption, the Markov model also assumes discrete transition time intervals, constant bridge population and stationary transition probabilities [10].

The transition probabilities obtained are represented by a transition probability matrix (P) of dimension n×n where each element $p_{ij}$ shows the probability of the condition of the bridge moving from state i to state j during a single time-step where n is the number of condition states.

In order to make calculations easier, a common simplification can be made. Noting that without intervention the bridge cannot be improved, therefore the assumption can be made that all elements in the matrix where i>j are equal to zero [11]. In addition to this, the final term in the matrix $p_{nn}$, where n is the number of states, is equal to 1 because the condition cannot get any worse and therefore remains trapped in the state. This state is referred to as an absorbing state as it is impossible for the process to leave this state once it enters it [12]. So, in this context state n is an absorbing state because the process is unable to leave this state without intervention.

Throughout the literature, this matrix has been further simplified to ease computation. This simplification comes from assuming that the condition of the bridge cannot deteriorate more than one step between inspections, therefore, when j>i, the value $p_{ij}$ is set to zero. Furthermore, since each row must sum to 1, once $p_{ii}$ is determined, the other values in the matrix can be determined by $p_{i,i+1}=1-p_{ii}$. Taking a four-state system as an example, these simplifications can be shown through the transition probability matrix shown in Equation (2) below.
(2)$$P=\left[\begin{array}{cccc} p_{11} & 1-p_{11} & 0 & 0\\ 0 & p_{22} & 1-p_{22} & 0\\ 0 & 0 & p_{33} & 1-p_{33}\\ 0 & 0 & 0 & 1 \end{array}\right]$$

Even though this simplification is attractive as it reduces computational complexity, it does have a major drawback. Not including the possibility of the bridge deteriorating by more than one state between inspections excludes the use of data where this has been the case, and the accuracy of predictions of future performance is reduced as not all possible states are considered.

Once these probabilities are calculated, the matrix can be used to predict the future performance of the bridge. The future condition vector can be obtained by using Equation (3) below.
(3)$$P\left(t\right)\;=P\left(0\right)\times P^{t},$$
where $P\left(t\right)$ represents the condition state vector at any number of transition periods t, $P\left(0\right)$ represents the initial condition of the bridge and $P^{t}$ represents the transition matrix raised to the power of the number of transitions.

#### Methods for Calculating Transition Probabilities

Throughout the literature, there are many methods that have been used to calculate infrastructure transition probabilities. Two of the most commonly used methods are a regression-based optimization method which is better known as the expected value method and the percentage prediction method.

First, the regression-based optimization method starts with sub-dividing the data into groups based on age in order to preserve the homogeneity assumption since the deterioration rate of a bridge is not the same at different ages [10]. Within each group, linear regression is performed based on the condition state being the dependent variable and the age of the bridge as the independent variable. This leads to the estimation of the transition probabilities through minimizing the absolute difference between the expected value of the condition state predicted by the regression model and the estimated value of the condition rating by the Markov chain, see Equation (4) below.
(4)$$\min\overset{N}{\underset{t=1}{\sum }}\left|A\left(t\right)-E\left(t\right)\right|\;\mathrm{subject}\;\mathrm{to}\;0\leq p\left(i\right)\leq 1,i=1,2,\ldots ,U$$

In Equation (4), N is the number of years in one age group, $A\left(t\right)$ is the average condition rating at time t based on the regression function, $E\left(t\right)$ is the estimated value of the condition at time t based on the Markov chain and U is the number of unknown probabilities.

There are many limitations to this method. For example, the use of linear regression to model the condition rating based on the age of the bridge is not suitable as the dependent variable is discrete and ordinal. In addition to this, the assumption of zero mean and constant variance is not met [13]. Furthermore, the partition of the data performed in the first step of the method results in smaller sample sizes which leads to less reliable results while also having the potential of introducing bias. Another drawback of this method is that the non-linear optimization cannot handle cases when the condition ratings in a certain age group either remain constant or increase instead of decrease as expected, thus resulting in the unity matrix (or close to it) [14].

The second method, which is most commonly used, is called the percentage prediction method. The probabilities are estimated based on Equation (5) below, where $n_{ij}$ is the number of transitions from state i to state j within a given time period and $n_{i}$ is the total number of bridges in state i before transition.
(5)$$p_{ij}=\frac{n_{ij}}{n_{i}}$$

In order to generate reliable and accurate transition probabilities, this method requires a large amount of inspection data from bridges with at least two consecutive condition records without maintenance interventions [6].

### 2.2. Deterioration Model in the Current BMS

Despite the shortcomings discussed in the previous section, the Markov model is embedded within the two bridge management systems known as Pontis and BRIDGIT. These two systems are widely regarded as the most advanced in bridge management. The use of Markov chains to model the deterioration leads to the use of a Markov Decision Process (MDP) to produce an optimal maintenance policy. The MDP links the current bridge condition and maintenance actions to the future condition of the bridge by using the calculated transition probabilities [9]. Pontis and BRIDGIT use the percentage prediction method (described in Section 2.1) in conjunction with expert elicitation. This allows for probabilities to be calculated and then adjusted based on the opinions of experienced engineers. In addition to this, Bayesian updating [15] is used to update these probabilities as new inspection data become available. One transition matrix is calculated for the entire bridge stock and is used for the entire lifespan of the bridge. One assumption that is made when using one transition matrix for the entire lifespan of the bridge is that the deterioration of the bridge is the same at any age. For example, a bridge which is 10 years old has the same probability of staying in the same state or transiting to the next worse condition state as a bridge that is 100 years old.

### 2.3. Improvements to the Markov Model

Due to the many limitations and assumptions of the previously described special case Markov chain, some research has focused on improving its accuracy. Two of these attempts are discussed here. Firstly, using Bayes’ theorem to adjust for the variation in the inspection period and secondly an Artificial Neural Network (ANN) is introduced with the aim to provide more reliable and accurate data.

The variation in the inspection period is evident in the case study data presented in Section 3, where it was found that over 90% of inspections are carried out between 1 and 3 years. This variation in the inspection can be accounted for by using Bayes’ theorem. This is achieved by multiplying the unadjusted transition probability by an adjustment factor as shown in Equation (6). For a given inspection period $\left(IP=X\right)$, the conditional probability of transition $\left(T\right)$ from state i to state $j\;\left(T=i\to j\right)$ is given by Equation (6).
(6)$$P\left(T=i\to j|IP=X\right)=P\left(T=i\to j\right)\times \frac{P\left(IP=X|T=i\to j\right)}{P\left(IP=X\right)}$$

Morcous [6] reaches the conclusion that adjusting for the variation in inspections improves results because not accounting for the gap in inspection can result in an error of up to 13 years when predicting the service life of a bridge.

Lee [16] made the point that one of the main difficulties that is faced when implementing deterioration models is that lack of reliable and accurate data. This can lead to unreliable prediction of future condition. The authors aim to address this issue by introducing a method to generate historical condition ratings. This proposed model uses an Artificial Neural Network-based Backward Prediction Model (ANN-BPM). The backward prediction allows for bridge condition ratings for a specific time period to be predicted. The first step requires determining relationships between existing condition ratings and non-bridge factors. The non-bridge factors used are those that either directly or indirectly effect the deterioration rate such as climate, traffic and population. These correlations are then used by the neural network to generate the historical information. In order for this method to have maximum benefit, the choice of non-bridge factors is crucial as the addition of relevant factors will lead to a more accurate prediction. This identification of factors can be achieved by performing parametric studies to determine the most appropriate factors and the optimum number of factors that should be used.

This method was used by Guoping Bu [10] along with a Markovian-based deterioration model. While the authors noted that this method did lead to more accurate prediction, they concluded that the use of this method did not address the problems that arise from the many assumptions and limitations of the previously described Markov model.

### 2.4. Semi-Markov Model

Using a semi-Markov process to model bridge deterioration was introduced by Ng and Moses (1998) [12]. For the semi-Markov process, as with the Markov process, the probability of moving from state i to state j is defined by the transition probability $p_{ij}$. Before transition to the next state the process will stay in the current state i for a duration $T_{ij}$. This time is known as the holding or sojourn time which is defined by the holding time probability density function $h_{ij}\left(t\right)$. The bridge will begin in the best state, remain there for a period of time before transitioning to the next worse state and so on. The Markov model is a special case of the semi-Markov model where the holding time distributions are geometric and exponential, respectively. The use of these distributions is restrictive and using the more generalized semi-Markov model means these restrictions are relaxed.

Several methods have been utilized to determine the holding or sojourn times. Firstly, in [12] where it was shown that if the time to reach each state $T_{i}$ can be determined, then the distribution for holding times $T_{ij}$ can be found by taking the difference between $T_{i}$ and $T_{j}$. By considering the difference between two arbitrary random Z=X−Y, the probability density function (pdf) of Z in shown in Equation (7) below.
(7)$$f_{Z}\left(z\right)=\int_{-\infty }^{\infty }f_{X,Y}\left(x,x-z\right)\mathrm{dx}=\int_{z}^{\infty }f_{Y}\left(x-z\right)\cdot f_{X|Y}\left(x,x-z\right)\mathrm{dx},$$
where $f_{X|Y}\left(x|y\right)$ is the conditional distribution of X given Y. By making an assumption that the number of transitions that occur at a point in the history of the bridge deterioration process follows a quasi-Weibull process, then the pdf of Z can be determined [12], where both *Y* is Weibull distributed with parameters $\left(\alpha_{1},\beta_{1}\right)$ and *X* is Weibull distributed with parameters $\left(\alpha_{2},\beta_{2}\right).\;$By letting $Y=T_{i}$ and $X=T_{i+1}$ then the pdfs of the holding time $h_{i+1}\left(t\right)$ for i=1,2,…,N can be computed. Once the holding time distribution has been obtained, it can be used to determine the interval probability matrix $\left\{\psi_{ij}\left(t\right)\right\}\;$which describes the probability that the process will be in state j at time t given that the process was in state i at time zero and is required for performance prediction, see Equation (8) below.
(8)$$P\left(t\right)=P\left(0\right)\times \left\{\psi_{ij}\left(t\right)\right\},$$

Another method to determine the holding time distribution is based on time-dependent reliability theory or survival analysis. Time-dependent reliability theory was introduced to describe the deterioration duration time within an individual condition [17]. This is achieved by defining the non-negative random variable T to be in the time that is spent in a specific condition rating. The cumulative distribution function $F_{i}\left(t\right)$ is shown in Equation (9) along with its relationship to the corresponding probability density function $f_{i}\left(t\right)$.
(9)$$F_{i}\left(t\right)=P\left(T\leq t\right)=\int_{0}^{t}f_{i}\left(x\right)\mathrm{dx}$$

The reliability or survivor function [18] can be defined as shown in Equation (10). This function defines the probability that the bridge will still be in condition state i by time t.
(10)$$S_{i}\left(t\right)=P\left(T>t\right)=1-F_{i}\left(t\right)$$

In addition to this, the hazard function or failure rate is defined as the instantaneous rate of failure and can be shown in Equation (11). Failure in this context means the bridge transitions out of the current condition rating.
(11)$$h_{i}\left(t\right)=\frac{f_{i}\left(t\right)}{1-F_{i}\left(t\right)}=\frac{f_{i}\left(t\right)}{S_{i}\left(t\right)}$$

For the discrete time case, the probability of transition out of state i to any lower state j within a period Δ after time t, is defined by Equation (12) below. Moreover, the probability of remaining in the current state i is defined by Equation (13). In order to allow deterioration by more than one state, an extension of these formulae was derived in [5].
(12)$$p_{ij}\left(t,\Delta \right)=P\left(t<T<t+\Delta |T>t\right)=\frac{F_{i}\left(t+\Delta \right)-F_{i}\left(t\right)}{S_{i}\left(t\right)}=1-\frac{S_{i}\left(t+\Delta \right)}{S_{i}\left(t\right)}$$
(13)$$p_{ii}\left(t,\Delta \right)=1-p_{ij}\left(t,\Delta \right)=\frac{S_{i}\left(t+\Delta \right)}{S_{i}\left(t\right)}$$

Many different functions have been investigated including Weibull and lognormal. In many cases, the Weibull distribution has been preferred because of its flexibility in fitting a variety of different kinds of data. The relevant functions for the Weibull distribution are shown in Equations (15) to (18) in Section 2.5.

### 2.5. Survival Analysis

Survival analysis is a statistical approach that allows the modelling of time to event data along with their associated contributing covariates. Survival analysis has been used to model the time spent in each condition state and to determine the service life of bridges. The two main functions associated with survival analysis are the survival and hazard functions which are defined in Equations (10) and (11). One of the main advantages of using survival analysis compared to other models is that it can handle censored observations. There are two main types of censoring: right censoring and left censoring [19]. Right censoring occurs when the bridge condition rating has not reached a certain value at the time of the bridge inspection. Alternatively, left censoring considers the case when the bridge has already surpassed the rating at the time of inspection. Censoring is taken into consideration when the model parameters are estimated using the maximum likelihood method. For the corresponding likelihood and log-likelihood functions for the different types of censoring, see Tabatabai et al. [20].

There are three types of models considered, non-parametric, semi parametric and parametric models. Non-parametric models have been used on several occasions to model the time spent in each condition state. The most frequently used non-parametric method used is the Kaplan–Meier (K–M) method. This method was used by Beng et al. [21] where the authors stratified the data to look at the effect of several covariates on the survival curves. Yang [22] stated that while the use of non-parametric models is attractive because of their simplicity, the resulting hazard function may not be reliable since the shape of the hazard would change in the presence of covariates.

Semi-parametric models are often used as they overcome the aforementioned issues presented by the non-parametric models. These models make no assumption about the shape of the hazard function and they support multivariate analysis. Since there is no assumption made on the shape of the hazard this means that its shape solely depends on the data. One of the most widely used models is the Cox Proportional Hazards model [23]. The hazard function used in the Cox model is shown in Equation (14) below.
(14)$$h_{i}\left(t\right)=h_{0}\left(t\right)\exp\left[\beta_{1}X_{i}1+\ldots +\beta_{k}X_{ik}\right],$$
where $h_{i}\left(t\right)$ is the hazard for the ith individual and is the product of two factors, firstly $h_{0}\left(t\right)$ which is the baseline hazard function and secondly, a linear function of a set of k explanatory variables with β coefficients which are to be estimated.

Parametric models are widely used within infrastructure deterioration because of their advantages [22]. Since the shape of the hazard function is specified from the start it can be extrapolated into the future which can make it useful for predictive modelling and if the right distributional function can be chosen then in most cases parametric models produce more accurate results. Several functions have been investigated such as Weibull, log-logistic and lognormal. Throughout the literature, all the models were applied to the data and a statistical method, such as the likelihood ratio test or the Akaike Information Criterion (AIC), was used to select the most appropriate model. In most cases, the Weibull distribution was found to have the best fit. The relevant functions for the Weibull distribution are shown in Equations (15) to (18) below where $f\left(t\right)$ is the probability density function, $F\left(t\right)$ is the cumulative distribution function, $S\left(t\right)$ is the survival function and $h\left(t\right)$ is the hazard function.
(15)$$f\left(t\right)=\frac{\beta }{\eta }{\left(\frac{t}{\eta }\right)}^{\beta -1}\exp\left[-{\left(\frac{t}{\eta }\right)}^{\beta }\right]$$
(16)$$F\left(t\right)\;=\;1-\exp\left[-{\left(\frac{t}{\eta }\right)}^{\beta }\right]$$
(17)$$S\left(t\right)=\exp\left[-{\left(\frac{t}{\eta }\right)}^{\beta }\right]$$
(18)$$h\left(t\right)={\left(\frac{1}{\eta }\right)}^{\beta }\beta t^{\beta -1}$$

The parameters β and η are the shape and scale parameters, respectively. The value of β describes the failure rate over time where a decreasing failure rate occurs when β<1, a constant failure rate when β=1 and an increasing failure rate when β>1. Agrawal et al. [14] fitted the Weibull distribution to data from the state of New York and used the parameter estimates to determine other distributional characteristics such as the average time spent in each condition rating.

In addition to the Weibull distribution, Tabatabai et al. [24] introduced the Hypertabastic distribution for use in survival analysis. One important feature of this distribution is its ability to model many different hazard shapes. The author uses this distribution in later work to model deterioration in bridge decks firstly in the state of Wisconsin [20] then for all bridges in the 50 states and Puerto Rico [25] and finally in a study looking at the states in North America [26] where the emphasis was placed on studying the effect of de-icing salts. Furthermore, Nabizadeh et al. [27] used this distribution to look at deterioration in bridge superstructures in Wisconsin.

### 2.6. Artifical Intelligence (AI), Machine Learning and Data Mining Techniques

AI, machine learning and data mining techniques have been introduced as an alternative method for predicting the future condition of bridges as they address some of the assumptions and limitations of the Markov model. Cattan and Mohammadi [28] used Artificial Neural Networks (ANN) to determine the relationship between the subjective condition readings awarded at a time of inspection and various bridge parameters. The authors noted that there is a significant restriction in the use of neural networks because they are application specific. Therefore, using the same input data from a different bridge network could produce very different results. Tokdemir et al. [29] compared artificial neural networks and genetic algorithms in predicting bridge sufficiency ratings. This paper concluded that the genetic algorithms outperformed the artificial neural networks, however, the one disadvantage of the use of genetic algorithms is the lengthy training times required. While both of these models produced good results, it was noted that further work to improve results would include experimenting with the model parameters and the addition of a variable describing the maintenance history could improve results. Maintenance history is not readily available in most bridge management systems [6] and has been highlighted as a point of further research in many studies.

Two of the main drawbacks of the Markov model, which is used in bridge management systems, is the inability to include the effect of the bridge’s condition history on its future condition and the interactive effect of different deterioration mechanisms for bridge components. Morcous et al. [30] introduces a Case-Based Reasoning (CBR) model to address these limitations and shows proof-of-concept by applying this method to a data set of 259 concrete bridge decks. CBR is a memory-based process which solves the problem under investigation by using the solutions of similar past problems. This model assumes that two bridges will have a similar performance if they have similar structural features, environmental and operational characteristics, inspection and maintenance record. One of the most valuable characteristics of the CBR is its ability to improve its predictive power as new cases are added to the BMS.

In order to model the interactive effect of bridge elements, Rafiq et al. [31] uses a Bayesian Belief Network (BBN) to describe the relationship between elements of masonry arch bridges in the UK. The condition of the bridge is a function of the condition of the bridge’s major elements, such as the supports or the deck, and the major elements are dependent on the condition of the minor elements, for example, the abutment and parapet. This complex relationship is modelled through conditional probabilities. The authors also investigated an extension to the BBN known as a Dynamic Bayesian Network (DBN) which allows for a time-dependent model. It was revealed that this extension produced the capability of capturing a difference in the initial condition of the elements and a variation in deterioration rates.

Most recently, Martinez et al. [32] investigated and compared the performance of five classification models with varying complexity. These models were k-nearest neighbour, decisions tree, linear regression, artificial neural networks and deep learning networks. Before the models were fitted, extensive data cleansing and preparation was undertaken, and the steps taken were described in detail. Some of the steps taken included the discretization of BCI into three classes and converting all nominal attributes into binary. The tuned dataset which was used for the modelling contained data on the bridge’s characteristics, the current BCI inspection and previous two inspections. Each of the models was evaluated using several performance metrics, for example, the Root Mean Square Error (RMSE). The authors went on to discuss how budgeting for maintenance is based on the current BCI value with a different maintenance strategy assigned to each class. If the BCI value is close to the limits of its category, a small change can affect the maintenance required. Sensitivity analysis was considered for each of the models to determine how many bridges were deemed to be critical, i.e., the scheduled maintenance of the bridge was uncertain. After considering all of these factors, the paper concluded that the decision tree model was recommended for BCI prediction in the case study considered.

## 3. Case Study: Application to Visual Inspection Data from the Northern Ireland (NI) Road Network

The Department for Infrastructure (DfI) is responsible for the management and delivery of the road network in Northern Ireland. DfI owns and maintains over 6978 bridges varying in construction type, age, span and function. Figure 1 below provides an overview of the bridge stock. In summary, over 50% are of masonry arch construction with over 67% of bridges consisting of a cumulative span of less than 5 m. A total of 78% are single span, and the predominant function is road over river.

These bridges are subject to regular inspections as per the requirements set out in DB63/17, Volume 3 Highway Structures: Inspection and Maintenance document with the Design Manual for Roads and Bridges (DMRB). The inspection records date back to the early 1970s but were digitized in 1999 and evolved to adopt the Bridge Condition Index (BCI) in 2015. Prior to BCI, an inspection record detailed the condition score assigned to the bridge by the inspecting engineer, now referred to as a legacy inspection record. This was commonly a numerical value ranging between 1 and 4 where 1 indicates a structure with minimal defects and 4 indicates that immediate action needs to be taken. The wide range and uncertainty on the exact boundary of each category lead to inaccuracy in deterioration models because the condition ratings are broad and subdivided into only four categories. Likewise, each inspection was extremely subjective and within each condition rating there was no clear way of ranking the bridges in terms of priority for the allocation of funds preventing long term strategic planning. The introduction of BCI aimed to address this issue and facilitate a uniform national assessment rating for all bridges. During the last decade, advanced BMS began to gradually migrate inspections from the previous format to the BCI method. Ultimately BCI will facilitate significant improvement in the prediction of future bridge deterioration. In the short term, the lack of consistency between the methods means condition deterioration is no longer directly comparable over long periods of time and leads to uncertainty in the true condition of many bridges across strategic road networks. To address this issue in the current case study data, an exercise was undertaken to apply a BCI conversion rating to all legacy inspection records logged subsequent to January 2000, details have been previously published by the authors [33].

### Application of Survival Analysis Techniques

In order to apply the survival analysis approach to this data, the BCI values need to be discretized into four categories to represent the four condition ratings. These boundaries are shown in Table 1 below.

This section will outline how the Kaplan–Meier method described in Section 2.5. Has been applied to investigate the effect of covariates on survival in a particular condition rating. In this application, survival indicates that the bridge has remained in the same condition rating, and failure represents the bridge deteriorating to a worse state. By the definition of Time in Condition Rating (TICR), the observation is either uncensored (i.e., a complete observation) or right-censored [22]. Figure 2 shows the survival curves for condition ratings 1, 2 and 3 before transitioning to any worse condition rating. The survival curve for condition state 3 (represented by the blue line) shows a higher probability of remaining in condition 3 compared to condition state 2 (represented by the green line) and condition state 1 (represented by the red line). This figure shows that the time spent in each condition rating state increases as the condition worsens.

K–M curves can be stratified to look at the effect of different categorical covariates on the survival. Looking at time in state 1, the following characteristics were considered. Firstly, if the bridge was masonry arch or not, secondly, the bridge’s function being road over river or not, thirdly if the bridge was single span or not and finally the class of road. There are five different road class categories: M denotes motorways, A class roads are part of the strategic road network often dual carriageways but not motorways, B class are important roads but not the trunk roads, C roads are smaller again and U represents unclassified roads such as rural roads. The stratified K–M survival curve for each of these characteristics is shown in Figure 3a–d, respectively. From Figure 3a, it would suggest there is a difference in the time spent in condition state 1 for masonry arch and non-masonry arch bridges. A hypothesis test can be performed to determine if this difference is statistically significant or not. The Wald’s test is used to test the null hypothesis that there is a statistically significant difference in the survival between the groups and the alternative hypothesis states there is no significant difference. In this case, the *p*-value was less than 0.05, which suggests that there is sufficient evidence to reject the null hypothesis in favor of the alternative at the 95% level, meaning that there is a significant difference in the time spent in condition state 1 for masonry arch and non-masonry arch bridges.

Figure 3d suggests a significant difference between motorways (M class) and all the other road classes. This can be attributed to the contractual obligation of a third-party organization to maintain bridges within the NI motorway network under Design Build Finance and Operate (DBFO) contracts.

A similar procedure was carried out to determine if the other characteristics show a significant difference in time spent in condition state 1. The results for the significance test are shown in Table 2 below. In addition to this, the effect of these covariates on the time in state for condition state 2 and 3 is also shown in Table 2.

By using these variables in a Cox Proportional Hazards model, the regression coefficients tell us the effect these variables have on the hazard and in turn on survival. The baseline bridge is one that is multi-span, not a road over a river, not masonry arch and on an unclassified road. The coefficients are shown in Table 3 below as well as the exponential of these coefficients which provides the hazard ratio. A hazard ratio equal to 1 represents no effect on the hazard, a hazard ratio of over 1 is associated with an increase in the hazard of the event and a decrease in the hazard is shown by a value below 1. Taking the masonry arch bridge as an example, the hazard ratio of 1.77 means that a masonry arch bridge is 1.77 times more likely to deteriorate to a worse condition from condition state 1 than a non-masonry arch. If the confidence interval for the exponential of the coefficient contains 1 then there is no effect of this covariate on the hazard. From the confidence intervals shown in Table 3 below, it is evident that a road class of M, the bridge being a road over a river and it being a masonry arch all suggest an effect on the hazard since their respective confidence do not contain 1.

This model was applied to the time spent in condition state 2. One notable hazard ratio from the results was the hazard of a masonry arch bridge deteriorating to a worse condition state from state 2 compared to a non-masonry arch bridge which is 0.799. The reason behind this change in hazard ratio will be discussed in Section 4.

## 4. Discussion, Conclusions and Further Work

This paper shows an application of survival analysis to model the effect of bridge characteristics (covariates) on bridge survival time where the survival time represents that spent in each condition state of the bridge. This has been implemented for all bridges in the Northern Ireland road network. Results from the K–M survival curves show that the time spent in the earlier condition states is lower that the time spent in the later condition states. The time in each condition state was analyzed separately to look at the effect of particular factors; these effects were further examined by applying the Cox Proportional Hazards model which gives hazard ratios. The high proportion of masonry arches in ageing transport systems increases the complexity in developing accurate predictive models for network-wide decision making.

As outlined in Section 3, the increased hazard ratio at the early stages of deterioration compared to later stages highlights the significant variation in performance of such structures relative to non-masonry arches. This is likely attributed to by the presence of corrosion or cracking in concrete or steel structures, which would be more obvious to the inspecting engineer. The results indicate that once initial deterioration of masonry arches occurs the rate stabilizes providing better long-term performance. Contrary to this, concrete/steel structures are likely to deteriorate more rapidly once initial signs of cracking/corrosion appear in the structure.

SHM data can provide valuable insights on the overall performance of concrete and steel in normalized environmental conditions and therefore have the potential to enable PBSHM for network-wide performance. Recent advances in standards relating to rebar cover and increased quality control during the construction stage should have measurable improvements on the future performance of concrete structures. It will be a number of years before this can be validated as approximately only 1.6% of the bridge stock in NI is less than 10 years old. Overall the age of bridges within UK road networks is significantly different to the relatively new structures used to develop predictive models from data held by the US Department of Transportation Federal Highway Administration (FHWA) National Bridge Inventory (NBI). The literature has shown significant advances in the development of predictive analysis models based on the NBI data. This has been made possible through the open access to all bridge data held on the system. The database provides detailed information on all public road bridges greater than 20 feet (~6 m), which currently totals 617,084 bridges. The FHWA provides a clear uniform method of inputting condition and assessment data to the system via specification documents provided on the portal. The system enables state and national level analysis on the condition and performance of the national bridges which subsequently informs future investment strategies. This vast dataset provides the opportunity to identify trends in bridge performance with more reliable results. Over 80% of bridges in the NBI have concrete deck construction and deterioration of this element is the most common factor in a bridge obtaining a poor condition rating. The deck performance is often isolated to provide a homogenous data set for the development of predictive deterioration models. Significant challenges exist in the adoption of such models to heterogeneous bridge stocks within European and UK bridge networks. The primary limitation is the lack of a widely adopted bridge inventory system and the associated extent of historic bridge condition data available, coupled with the age of structures within UK networks. Over 77% of bridges within the NI database are greater than 50 years old compared with only 40% of bridges held on the NBI. As a result, a large proportion of NI bridges will have been subject to maintenance at some point. This study has not looked at the effect of maintenance on the time in each condition state and this will be the focus of further work. The data show several occasions of an increase in condition state which can only be assumed to have occurred due to rehabilitation or replacement.

There were four factors investigated which were the bridge construction type, function, number of spans and the road class. The database contained other construction information about the bridges including deck width, skew angle and cumulative span but these variables were not included in this study due to large amounts of unavailable data. In addition to this, the variable which indicates the age of the bridge at the time of the change of condition state was not included in the models as it was highly correlated with the construction type of the bridges. This shows that these covariates are not independent of each other and only one of them should be used in the analysis [27].

The data introduced in Section 3 show that there are many differences between the bridge stock and inspection data from the United States and Northern Ireland. The BCI value used in this study is based on the BCI average value which represents the condition of all the elements of the bridge. All inspections are also given a BCI critical value which represents the condition of the bridge elements which are deemed to be of very high importance to the bridges function. Any research which uses BCI as an indicator of deterioration focuses on BCI average, however, not using the BCI critical results in a loss of information, therefore a point of future research will look at incorporating both of these values into deterioration models.

## Figures and Tables

**Figure 1 sensors-20-06894-f001:**
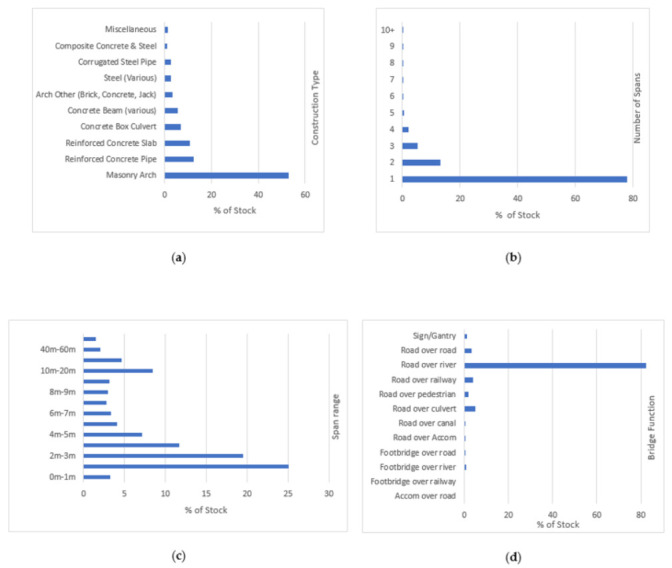
Overview of bridge stock on Northern Ireland road network: (**a**) span construction type; (**b**) number of spans; (**c**) cumulative span range; (**d**) bridge function.

**Figure 2 sensors-20-06894-f002:**
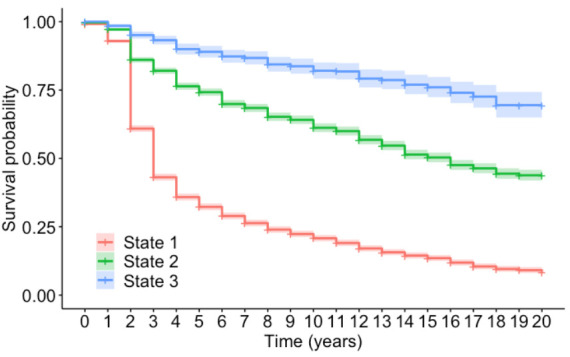
A graph showing the survival curves for time-in-state 1, 2 and 3.

**Figure 3 sensors-20-06894-f003:**
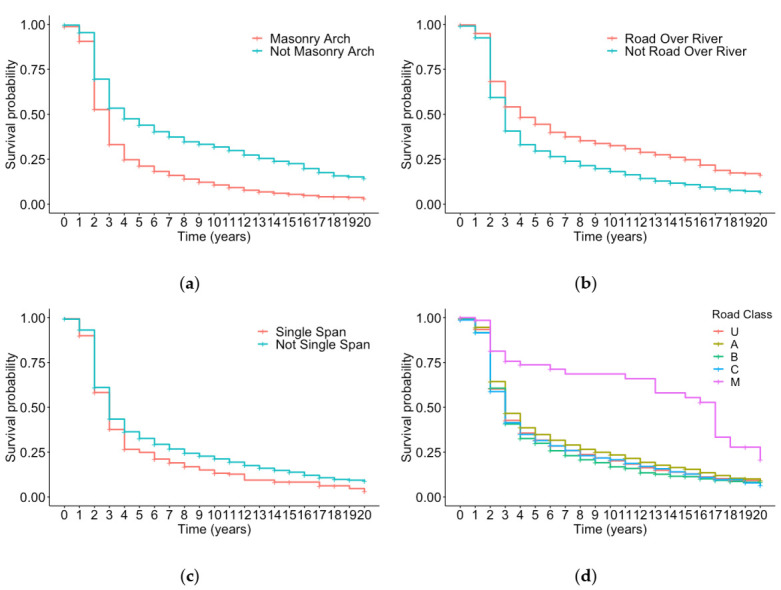
Kaplan–Meier survival curve for time spent in condition state 1 stratified by: (**a**) the bridge being masonry arch or not; (**b**) the bridges function being road over river or not; (**c**) the bridge being single span or not; (**d**) the road class.

**Table 1 sensors-20-06894-t001:** Bridge Condition Index (BCI) boundaries for the four condition ratings.

Condition Rating	BCI Boundaries
1	[83,100]
2	[73,83)
3	[53,73)
4	[0,53)

**Table 2 sensors-20-06894-t002:** A table showing the results of the hypothesis test for each of the characteristics for the time spent in condition states 1, 2 and 3 where * indicates significance at 5% level, ** highly significant, *** very highly significant and × denotes the test was insignificant at the 5% level.

Bridge Characteristic	State 1	State 2	State 3
Masonry Arch and Not Masonry Arch	***	×	×
Road Over River and Not Road Over River	***	∗∗∗	×
Single Span and Not Single Span	∗	×	×
Road Class	∗∗∗	∗∗∗	×

**Table 3 sensors-20-06894-t003:** A table showing the value of coefficients, the exponential of these coefficients and the confidence intervals of the exponential of the coefficient for the significant variables in Cox Proportional Hazards model for time in condition state 1.

Variable	Coefficient (to 3sf)	Exp(coef) to 3sf	Confidence Interval for Exp(coef)
Single Span	−0.125	0.883	[0.778,1.00]
Road Class—A	0.0822	1.09	[0.998,1.18]
Road Class—B	0.0490	1.05	[0.967,1.14]
Road Class—C	0.0143	1.01	[0.94,1.09]
Road Class—M	−0.570	0.566	[0.402,0.797]
Road over River	0.102	1.11	[1.02,1.21]
Masonry Arch	0.572	1.77	[1.66,1.89]

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
