# Peer review of "Identification of Bridge Key Performance Indicators Using Survival Analysis for Future Network-Wide Structural Health Monitoring"

_sensors, 2020, doi:10.3390/s20236894_

Round 1

Reviewer 1 Report

From the perspective of bridge networks, this paper presents a framework for bridge rating. In the paper, the previous literature has been reviewed. Survival probability technique has been used to model the time spent in each condition rating. This study is attractive. In case study (section 4), the reviewer found no application of structural health monitoring. This is what the reviewer is confused with.

Author Response

The authors wish to thank you for taking the time to review our manuscript and for providing such helpful comments. Please see attached document for our response. We hope this addresses your concerns relating to our paper.

Reviewer 2 Report

The paper focuses on an important topic: the management of the bridge networks. It can be improved from the following aspects.

  1. The title is very broad and not specific to this research. It is suggested to change the title to represent the main contribution of this paper(survival analysis?).
  2. The structure of this paper is a little bit confusing, as it shows the review in Section 3. It is suggested to re-organise the order of the sections.
  3. Survival analysis is a statistical approach, which is not proposed by the authors. So what is the contribution of this paper?
  4. It is not clear how the survival analysis is verified based on the data. 
  5. Are there any other options? Why survival analysis is selected? 

Author Response

(The authors gave the same response as above.)

Reviewer 3 Report

The paper is very well written and is a comprehensive data analysis of the bridges and their condition deterioration.

The major issue with the paper is that it does not fit well into the frame of Sensors, as there is no sensor data or signal processing involved. So the reviewer will recommend some other journals which will be more suitable.

The other issue is the paper states that it reviews the state of the art, but only a few papers are reviewed in detail, as the paper is a discussion and review more papers need to be reviewed. There is a lot of work in the use of AI for resource management in the offshore structures, both oil and gas and offshore wind, which could be of interest.

Author Response

(The authors gave the same response as above.)

Reviewer 4 Report

The paper presents a review of Markov theory and its application to visual inspection data from road network in Northern Ireland.

The paper is well written and informative, but sensors seem to have no role in the process. In addition, the novelty of the study is limited and no new knowledge is developed, so the manuscript can be accepted as a review paper rather than as an original article, provided that minor typos and misprints are corrected throughout the paper.

Author Response

(The authors gave the same response as above.)

Round 2

Reviewer 3 Report

The paper is a slightly improved version of a previous manuscript.

The paper is well written and has useful information for the readers.

The issue with the paper is:

It claims that it reviews the literature, but only a few papers have been reviewed. Furthermore, if it is a review paper more papers from the last 5 years need to be discussed. There is a lot of work on AI in the last 5 years which can be discussed. 

Also, the section 2 which claims to be a review is more a description of methodology rather on the careful analysis and critique of the literature. If the authors claim that one one the highlights of the paper is the literature review. More discussions on the contributions of the paper need to be carried out.

So either the claim of the paper of the review needs to be omitted or the whole section needs to be considerably improved.

Author Response

Thank you for you for taking the time to review the revised version of our paper and for providing very helpful comments. The authors agree there has been many advancements in AI solutions for bridge management in recent years. We have selected a number of options which may be applicable to a European road network and acknowledge that this does not cover everything. To address this in the paper we have updated the abstract and introduction to clarify the contribution of our paper.

The paper does not provide a comprehensive description of all maintenance methods and policies available in the literature. A critical review of some of the main approaches in bridge predictive maintenance modelling is presented and the challenges in their adaptation to the future network wide management of bridges are outlined.

We have also updated the title of section 2.

Methodologies adopted for current bridge predictive maintenance methods